# Evaluation and Comparison of Ultrasonic and UWB Technology for Indoor Localization in an Industrial Environment

**DOI:** 10.3390/s22082927

**Published:** 2022-04-11

**Authors:** Amalia Lelia Crețu-Sîrcu, Henrik Schiøler, Jens Peter Cederholm, Ion Sîrcu, Allan Schjørring, Ignacio Rodriguez Larrad, Gilberto Berardinelli, Ole Madsen

**Affiliations:** 1Department for Materials and Production, Aalborg University, 9220 Aalborg, Denmark; lelia.cretu@gmail.com (A.L.C.-S.); sircu.ion@gmail.com (I.S.); om@mp.aau.dk (O.M.); 2Department for Electronic Systems, Aalborg University, 9220 Aalborg, Denmark; alsc@es.aau.dk (A.S.); irl@es.aau.dk (I.R.L.); gb@es.aau.dk (G.B.); 3Department for Planning, Aalborg University, 9000 Aalborg, Denmark; pce@plan.aau.dk

**Keywords:** indoor localization, TDoA, radio based, ultra-sonic, industrial, mobile robots, evaluation, comparison

## Abstract

Evaluations of different technologies and solutions for indoor localization exist but only a few are aimed at the industrial context. In this paper, we compare and analyze two prominent solutions based on Ultra Wide Band Radio (Pozyx) and Ultrasound (GoT), both installed in an industrial manufacturing laboratory. The comparison comprises a static and a dynamic case. The static case evaluates average localization errors over 90 s intervals for 100 ground-truth points at three different heights, corresponding to different relevant objects in an industrial environment: mobile robots, pallets, forklifts and worker helmets. The average error obtained across the laboratory is similar for both systems and is between 0.3 m and 0.6 m, with higher errors for low altitudes. The dynamic case is performed with a mobile robot travelling with an average speed of 0.5 m/s at a height of 0.3 m. In this case, low frequency error components are filtered out to focus the comparison on dynamic errors. Average dynamic errors are within 0.3–0.4 m for Pozyx and within 0.1–0.2 m for GoT. Results show an acceptable accuracy required for tracking people or objects and could serve as a guideline for the least achievable accuracy when applied for mobile robotics in conjunction with other elements of a robotic navigation stack.

## 1. Introduction

With the introduction of the fourth industrial revolution [1] the need for mass customization and, in turn, massive flexibility of manufacturing facilities has been accentuated. Manufacturing flexibility may be achieved in at least two ways: reconfigurable flow topology, i.e., conveyors/processing stations, and on the other hand, autonomous mobile robots (AMR) for transport of products and processing apparatus. Essential to mobile robotics are the three fundamental questions for reactive mobility [2]: Where Am I?, What Surrounds Me?, What Should I Do Next? The first question has been investigated for outdoors cases such as intelligent (precision) farming [3], where the availability of global navigation satellite systems (GNSS) such as GPS has constantly improved over the last decade. For indoor (GPS-denied) cases, the development has been towards independence from navigational infrastructure, i.e., navigation stacks based on robot sensors only. Wheel and visual odometry along with LIDAR and/or RADAR allow non-drifting relative positioning within a neighbourhood of the initial robot pose, whereas range may be extended through Simultaneous Localization and Mapping (SLAM) [4] at the expense of drifting until loop closure. Whereas odometry and SLAM provide high levels of short-range precision, infrastructural localization provides lower precision at a higher range. In other words, infrastructural localization removes drift from odometry methods [5,6], whereas odometry complements infrastructure with precision. The advantage of augmenting odometry with the installation of infrastructure obviously depends on the precision it offers. We evaluate and compare, with respect to accuracy, two prominent candidates for indoor localization infrastructure—radio-based (RF) and ultrasound-based (US).

Indoor positioning technologies try to solve the problem of a sufficiently accurate positioning in various applications and environments. The main applications for indoor positioning systems (IPS) are: tracking of people, robots and other objects in diverse environments, such as offices, hospitals, warehouses, production halls, barns, etc. The authors of [7] mention that the industrial environment is proven to steadily adapt indoor positioning systems for the availability of positioning data which is a valuable source of information for process automation and optimization. Indoor localization technologies have aspects which should be considered on an application basis in terms of required accuracy. The challenge of obtaining accurate positioning comes from different sources found indoors: dynamic environments, obstacle density and materials type. These sources create disturbances for the IPS: Non-Line of Sight (NLoS), shadowing, multi-path, interference, etc. with a significant diversity among environments [8].

### 1.1. Related Work and State of The Art (SOTA)

A plethora of technologies and combinations for improving position accuracy are presented and evaluated in the literature. Technologies are RF-based (cellular, WLAN, ultra-wideband (UWB), Bluetooth), image detection–based (infrared (IR), optical, LiDAR, RADAR cameras) or sound-based (acoustics and ultrasound (US)). Several past studies have performed surveys on the landscape of indoor positioning systems at the time [9,10,11,12,13,14]. Few of these studies propose specific parameters that help with evaluating indoor positioning systems in an attempt to draw general conclusions about a technology. The authors of [8] mention that throughout this field of research IPS are evaluated in particular environments resulting in conclusions that cannot be generalized, nor compared to other systems’ results. Although various metrics are proposed in the literature for parameterizing environmental characteristics, there are variations in the real world which are difficult to capture. The authors of [12] argue that when choosing between technologies, the parameters should match the application requirements, otherwise a trade-off is designed. The works in [9,11] suggest a taxonomy of features for evaluating systems. Different parameters proposed by [9,11] are used in this study to evaluate the two systems. The parameters used for comparison are accuracy, coverage, scalability, cost and type of environment.

The most popular wireless signals used in device ranging are sound-based and radio frequency (RF). Within sound-based systems, there has been an increase in ultrasound positioning technologies starting with systems such as the one presented in [15]. In the RF domain, positioning systems based on Wi-Fi and Bluetooth flood the literature—recently popularizing Bluetooth Low-Energy (BLE) and UWB [8,16]. RF-based technologies gained traction due to their adaptability to different use cases [7]. With RF signals travelling at the speed of light, penetration characteristics give them an advantage over US signals that require Line of Sight (LoS) due to low propagation speed [16]. Based on accuracy results obtained across studies in the literature, BLE-based systems are viewed as proximity technologies compared to UWB systems which are generally more accurate, but recent improvements involving Angle of Arrival (AoA) approaches have moved the accuracy of BLE closer to that of UWB.

Though studies investigating commercial UWB-based technologies in an industrial-like environment are still scarce, a growing trend can be observed as the industry uptake of such technology rises. The authors of [7] evaluated the Pozyx Creator system in an industrial warehouse with metallic obstacles in conditions of LoS and NLoS. Pozyx [17] is based on the same DecaWaveDWM1000 chip as ATLAS in [18]. This study also aimed to improve the Pozyx proprietary positioning algorithm “Pozyx_POS_ALG_UWB_ONLY” with a multilateration algorithm. The results are based on measurements of nine ground-truth points taking 20 samples per point. The (by Pozyx) claimed accuracy of 0.1 m could be verified under LoS conditions [17] but experiments showed an average error under mixed conditions (LoS/NLoS) of 0.862 m. The standard deviation lies between 0.014 and 1.525 m depending on the area. The multilateration algorithm showed improved results of 0.51 m mean error, a 40% improvement compared to Pozyx’s algorithm. These were sufficient conditions to evaluate and contest Pozyx-claimed accuracy. The present study employs the Pozyx technology, as well.

In [16], a study compared three UWB-based commercial technologies from Ubisense, BeSpoon and DecaWave. The experiment was carried out for static measurements, conducted in an industrial warehouse containing metal structures such as vehicles, robots, metallic shelves, etc. There are both LoS and NLoS conditions. Tags are tracked across 70 reference points with a 30 s sampling time per point. The accuracy results show the mean and median for the DecaWave is 0.49 and 0.39 m, BeSpoon is 0.71 and 0.58 m and Ubisense is 1.10 and 0.61 m (w/AoA method). The study did not include a dynamic experiment to compare the performance of the systems in mobility cases. The work in [18], however, investigates the dynamic performance of ATLAS using a mobile robot following a set trajectory. The ATLAS system is based on the DWM1000 module from Decawave. OptiTrack is used as a reference system. The experiment is performed in LoS conditions and repeated 10 times. The Euclidean error of the system shows an accuracy of 0.3 m in 2D and 0.456 m in 3D in the 99% quartile. This study presents dynamic measurements of the same UWB module used in [16] in geometrical LoS-only conditions for optimal testing conditions of the ATLAS system which are not similar to industrial environments conditions.

Using another UWB-based commercial technology rather than Pozyx, the authors of [19] evaluate the TimeDomain PulsON440 in a heavy industrial galvanic environment for plating fashion accessories with precious metals. The study mentions that such a system claims an accuracy up to 2 cm. The experiment uses four beacons and one tag, describing two static tests and one dynamic in conditions of both LoS and NLoS. The tag is placed on top of a metallic frame consecutively submerged in six tanks. The first static test in LOS conditions presents the best result as 0.02 m and the worst as 0.38 m mean error. The second test in NLoS conditions (operators present) shows deteriorated results with a mean error of 0.22 m. The dynamic case illustrates the trajectory of the frame across the six tanks, as moved by the operator, implying a case of both LOS and NLoS conditions; however, statistics are not described for this case. While the authors of the study claim the commercial system to be accurate enough, descriptive statistics are important to characterize a technology in different environments.

When it comes to proximity technologies, BLE is gaining momentum in the literature. Popular commercial systems based on BLE are iBeacons from Apple Inc. and EddyStone from Google Inc., providing proximity detection in the range of 1–3 m. Moreover, positioning is mostly perfomed in 1D using the received signal strength indicator (RSSI) as a parameter for distance [11]. The authors of [20] consider BLE a good candidate for accurate IPS claiming that a trade-off between BLE’s characteristics such as sampling time and energy consumption can provide the accuracy needed. Although there are many studies on BLE as an IPS [11,21], investigations stressing this technology in industrial or harsh environments are scarce. The authors of [22] perform a static and dynamic evaluation of BLE versus Wi-Fi using 11 beacons spread across the hallways of a building floor. In the static case, BLE outperforms Wi-Fi with an average Root Mean Square (RMS) error of 3.8 m versus 5.3 m. For the dynamic case, the signal is filtered with a particle filter (PF) and the Cumulative Distribution Function (CDF) indicates an RMS error of 4 m for BLE. The work in [23] finds that different filters yield different results with BLE. A PF yields an accuracy of 1.441 m, a cascaded Kalman filter (KF)-PF yields 1.03 m and cascaded PF-Extended KF yields 0.95 m [11]. The work in [24] also compares BLE results obtained using three positioning methods for a dynamic case: dead-reckoning, trilateration and KF. The best result was 0.73 m average accuracy using the filter method. The authors of [25] track multiple workers in a confined construction site with a BLE tag fusing RSSI with KF and an accelerometer. In the static case, the average error ranges from 0.2 to 0.22 m, and in dynamic cases, the average error ranges from 0.1 to 0.47 m. The work in [26] evaluates BLE in the harsh conditions of a public transport station with highly dynamic features and NLoS. The accuracy values are similar to the ones found in the literature for environments of an office or hallways, i.e., between 0.48 and 3.6 m. The authors of [27] study accuracy differences between Wi-Fi, BLE beacons and UWB tags in a 36 m^2^ classroom. The results show an average error of 1.39 m for Wi-Fi, 0.86 m for BLE and 0.24 m for UWB. Similar results were obtained in [28], comparing UWB and BLE and a fusion between the two. The medians of localization errors are 0.23 m for UWB and 1 m for BLE in a laboratory and elderly people’s home environments.

However, a growing number of studies are trying to achieve sub-centimetre accuracy results using methods such as fusion or Angle of Arrival/Departure (AoA/AoD). The works in [29] argue that with the arrival of new methods based on AoA/AoD, BLE can achieve more accurate measurements than those based on RSSI. The study carries static measurements of BLE 4.2 (old) and 5.0 (new) in an industrial laboratory: a classroom and a workshop. The workshop contains heavy machinery and other factory-like conditions. The best results obtained in the classroom for BLE5.0 range between ±2 m and a standard deviation of 1 m. The best results in the workshop for BLE5.0 are between 0.5 and 1.5 m of distance error. BLE4.2 showed an error between 0.5 and 1 m. In their study, the authors of [30] show that in LoS conditions in a sports hall, using AoA-only to determine distances to the device, the mean error is 0.04 m on the X-axis and 0.01 m on the Y-axis. It is noted again, as in previous studies, that as the distance between beacon and device increases, the error also increases. This error has also been shown to increase in the presence of disturbances due to multipath in NLoS conditions. AoA mitigation methods for fusing KF or IMU can be used to reduce errors. The work in [31] further investigates the feasibility of using the AoA and AoD information provided by BLE5.1 to determine the position of a device. The empirical study achieved an error of below 0.85 m for more than 95% of the positions. It shows that utilising AoA sub-meter accuracy can be achieved; however, there is still progress to be made to achieve centimetre-level accuracy.

Looking back at some of the known US-based technologies from the literature, it is obvious that systems such as BAT [32], Cricket [15] and Dolphin [10] have been used several times, e.g., in [9,10,11,12,13,33]. The Active BAT [32,34] from Olivetti and Oracle Research Laboratory (ORL) was supposed to meet the requirements of fine-granularity in office rooms which systems back then could not meet. In [11,14], the BAT system is mentioned as reaching 0.3 m in 3D. MIT’s Cricket system [15] is a location-support service by combining RF and US. The system is developed to be cheap, decentralized and provide no more than room-level precision. The authors of [11,13,33] mention the accuracy achieved by Cricket is 0.1 m in 3D and within 0.3 m 99% of the time. DOLPHIN [10]—Distributed Object Locating System for Physical-space InterNetworking—proposes to reduce configuration costs by deploying wireless sensor nodes that send and receive RF and ultrasonic signals (US). DOLPHIN has shown an accuracy of around 0.15 m using three beacons.

Novel positioning systems using US are described in [5,35,36,37,38] to name a few. In [11], the US technology is described as being able to provide fine-grained centimetre level accuracy. The work in [12] mentions that US provides an accuracy of 0.01 m with a medium-level cost and low complexity. The main disadvantages are represented by the accuracy limitation due to air temperature, humidity and pressure changes. In [36] is presented a US IPS based on a receiving node carried by the user and at least four or five transmitting nodes. It uses the Time Difference Of Arrival (TDoA) method and has an effective range of 10 m. The acclaimed accuracy of the system is 0.02 m in standard deviation. The authors of [37] developed and tested a robust indoor positioning system based on a US signal and a wireless sensor network. The measurements are evaluated against a commercial vision-based localization system with 1 mm precision. Using four anchors or beacons at a fixed position, the mobile robot moves in a circular trajectory at a very low speed in conditions of LoS. There are 4 tests carried out of different circle radii and the highest error of tracking the robot is 10.24 mm. The authors of [35] developed a low-cost FPGA-based 3D location system based on US. The location accuracy is 0.033 m in a static case. Furthermore, the system was tested on a robot used for point tracking. If the error was greater than 5 cm, the robot would reiterate the test until it reached the target point. Results showed that the robot reached the target point on the first try in 80% of cases, and in the remaining 20% of cases, it reached the target point in two attempts. Another study [5] proposes a low-cost IPS based on active US beacons for a mobile robot. US beacons are placed on the ceiling and two receivers are placed on the robot. The system measures the distance between the beacons and receivers directly. Using a Time of Flight (ToF) method for 2D positioning of the robot at a frequency of 10 Hz, the accuracy obtained is 40 mm with a standard deviation of 10 mm. Both studies [5,35] propose a new system setup for indoor localization, however, they are not thoroughly tested in NLoS or industrial conditions. A system based on the TDoA method and combining Orthogonal Frequency Division Multiplexing (OFDM) waveform with Zadoff–Chu sequences is also proposed in [39]. Performance analysis shows ranging accuracies in the order of 2 cm, with the excellent cross-correlation properties of the sequences ensuring large scalability.

In [38] is presented a comparative study between a US- and UWB-based indoor positioning systems. The systems are compared in a large indoor area (24 × 14 m or 336 m${}^{2}$). The US system uses a ToF method and the UWB system uses Round-Trip times (RTT) in conditions of LoS. The UWB system is a commercial product based on Decawave TREK1000 using three and six beacon nodes. The US system is provided by GEINTRA-US/RF based on their LOCATE-US prototype using three beacons. The comparison between the two systems is performed in different configurations: fixed positions, a mobile robot following a line, a person walking on a known path. In the static case, both systems presented a distance error of less than 0.2 m in 80% of cases in LoS conditions at a height of 1.5 m. The dynamic case involved a line-following robot with both systems tags on top tracking a 7.2 m long line at a height of 0.1 m. The CDF of the dynamic case showed an error of maximum 0.2 m in 80% of cases for the US system, a maximum of 0.12 m for the three-beacon UWB system and a lower than 0.08m error for the UWB system with six beacons. The test performed by a person walking a predefined path resulted in an error lower than 0.65 m in 80% of cases for the US system, whereas the results were 0.35 m for the UWB system with six beacons and an error of more than 0.5 m for the three-beacon system. When both systems have the same number of nodes, the errors are similar in movement scenarios.

A major challenge for both RF and US systems is a complex and dynamic environment due to noise, multi-path propagation and NLoS situations [13]. As noted before, a disadvantage of these systems is that their accuracy is dependent on the placement of the beacons on the ceiling or walls. Most methodologies in the literature for evaluating IPS consider LoS in environments that are less prone to interference, leading to results that cannot be generalized over multiple environments. Studies performing comparisons between different technologies in an industrial context in conditions of both LoS and NLoS are very few—see Table 1. Moreover, these methodologies present static measurements which are not realistic in an industrial context, or very precise trajectories followed by robots at low speeds.

In Table 1, the accuracy ranges of the technologies above are summarized. It can be noted that there are more studies performed on UWB in industrial settings than for US which predominantly are conducted in laboratories, similar to the BLE studies. The accuracy of UWB in industrial laboratories from selected SOTA ranges from 0.005 m for short ranges and LoS to 1.10 m as the worst performance. The accuracy of US in laboratories features an accuracy range of 0.001 m as the best performance to 0.1 m as the worst. The only industrial case for US shows an accuracy lower than 0.2 m in the best case and lower than 0.65 m for the worst case. The accuracy of BLE has been observed to range from 0.1 m to 3.8 m, thus further supporting the idea that BLE can be used as a proximity technology.

**Table 1 sensors-22-02927-t001:** SOTA Review and Summary.

System	Paper	Year	System	Comparison UWB vs. US	Room Size m${}^{2}$/ Environment	GT Points	LOS/ NLOS/ Mix	2D Accuracy [m] Mean ± Std
UWB	[40]	2014	802.15.4a compliant UWB System	No	5.3 × 11.5/ Office	5	LOS (Static P4)	< 0.4 ± 0.04
							LOS (Dynamic P4)	0.89 ± 0.08
							NLOS (Dynamic P4)	0.88 ± 0.1
	[18]	2016	ATLAS	No	Laboratory	8	LOS	0.21
	[16]	2017	BeSpoon	No	12 × 12/ Industrial Laboratory	70	Mix	0.71
			Ubisense					1.10
			DecaWave					0.49
	[7]	2019	Pozyx	No	Industrial Laboratory	9	LOS (1.5 m range)	1.5 ± 0.03
							NLOS (1.5 m range)	1.75 ± 0.03
							LOS (10.9 m range)	11.6 ± 1.7
							NLOS (10.9 m range)	11.6 ± 4.4
	[19]	2019	TimeDomain PulsON440	No	Galvanic Industry	6	LOS (Static)	0.38
							NLOS (Static)	0.22
	[41]	2019	Pozyx	No	Industrial Laboratory	ROS Simulation	LOS	0.22
							NLOS	>1
	[42]	2020	DecaWave	No	Industrial Laboratory	70	LOS (Static)	0.01 ± 0.01
							LOS (Dynamic)	0.21 ± 0.13
							Mix (Dynamic)	0.25 ± 0.09
US	[32]	1997	Active BAT	No	Office	-	LOS	0.03 [11,14]
	[5]	1998	Prototype	No	0.5 × 0.4	55	LOS (Static)	0.04 ± 0.01
	[15]	2000	MIT Cricket	No	Office	-	LOS	0.1 [13]
	[10]	2003	DOLPHIN	No	Office	-	LOS	
	[43]	2010	LOSNUS	No	Office	35	LOS (Static)	0.001
	[35]	2011	Prototype	No	1.2 × 1.8 m	20	LOS (Static)	0.03
	[36]	2016	Prototype	No	Laboratory	1	LOS (Static)	0.02
	[37]	2017	Prototype	No	Laboratory	-	LOS (Dynamic)	0.012
	[38]	2019	Decawave TREK1000 (UWB) Locate-US (US)	Yes	24 × 14/ Industrial Laboratory	5	Mix (Static)	<0.2 (UWB & US)
							Mix (Dynamic- robot)	<0.2 (US) <0.12 (UWB)
							Mix (Dynamic- moving person)	<0.65 (US) >0.5 (UWB)

### 1.2. Our Contribution

The value added to this field of research by our experimental 2D study is represented by:Comparison under the harsh conditions of an industrial environment between systems based on US and UWB;Large study on 100 ground-truth points and an uncontrolled environment;Measurements and analysis for both LoS and NLoS conditions;The involved methodology, including both static and dynamic cases inspired by industry: static localization for pallets and production modules, dynamic tracking of autonomous robots, forklifts, workers;Static measurements and analysis for different heights (0.3 m, 1 m, 2 m).

The innovation of our large measurement study stands with the extensive methodology and analysis chosen to describe the comparison. Such in-depth analysis is seldom seen in the literature for comparison studies. The analysis separates the results for static aggregate measurements across the 100 points—combining the effects of both LoS and NLoS, static single-point measurements in LoS-only conditions and dynamic measurements using filtering to reduce the static influences. We analyse the effects of obstacle density and height, as this can affect the accuracy of the tag measurements. In the static case, this is achieved by placing the tags at three heights on a tripod, as shown in Figure 1a. In the dynamic case, a mobile robot Figure 1b carries the tags and follows a trajectory through the laboratory in spaces without obstacles such as corridors, but also in-between machinery and obstacles. An involved analysis of the measurements is presented between the two cases using parameters proposed by previous studies [8,9,11,12]: accuracy and precision, cost, scalability and limitations of the technologies under assessment.

### 1.3. Layout

The layout of this paper starts with the introduction and related work in Section 1. Section 2 presents the technologies and solutions involved in the research including the reference system. This is followed by an overview of the laboratory facility, beacon- and ground-truth position–placement, as well as the measurement setup and methodology for static and dynamic cases. Hereafter, in Section 3, we present statistical results for the static case for 100 ground-truth positions at three different heights. Section 3.4 presents the statistical results for a laboratory wide trajectory traced by a small robotic vehicle. Section 4 discusses the results and conclusions along with directions for future work.

## 2. Materials and Methods

### 2.1. Localization Technologies

We investigate localization technologies that require no local signal processing. Thus, we exclude structured visual landmarks. The technologies provide indoor localization services similar to GNSS/GPS, i.e., where a real-time position estimate is offered in Cartesian coordinates. In this section, we present the technologies under study as well as a surveyor-quality system used for reference.

#### 2.1.1. UltraSonic (US)

We apply a development version from GamesOnTrack (GoT) [44] employing a combination of radio-based synchronization and US-based TDoA. Contrary to their legacy system, the development version applies a reverse protocol, i.e., fixed beacons act as transmitters, whereas mobile units act as receivers. Beacons transmit in a predefined pattern governed by a master station, where time slots may be reused on a distance basis. The non-simultaneous nature of transmissions leads to the necessity of combining TDoA information over time, which is performed by a patented projection algorithm. This may cause delay errors in the dynamic case, which is evaluated below. The reason for using the reverse system is to provide similarity with the UWB system in terms of scalability. It is assumed that scalability is most relevant in terms of the number of mobile units (tags), whereas fixed beacons follow the geometric proportions of the environment along with the possibility of time reuse. US bursts occupy the audio signal space for a longer duration than radio bursts, which means that the sampling/update rate of the legacy system decays as an inverse proportion of the number of mobile units. This is not the case for the reverse system, where sampling rate decays with the number of beacons within range. In the described case, we observe an effective update rate of 5 Hz.

#### 2.1.2. Ultra-Wideband (UWB) Radio

We apply a commercial product from Pozyx, which is based on the UWB TDOA technology base from DecaWave DW1000. The Pozyx system uses the Enterprise version which allows for unlimited number of tags to be tracked, variable update rates of up to 100 Hz and a claimed accuracy of 10–30 cm [17]. Pozyx beacons used in this study (see Figure 1a) represent an old hardware version which differs in specifications from the currently sold beacons. The Pozyx tags are the developer tags with a variable update rate of up to 100 Hz—see Figure 1b. Neither in the case of Pozyx, nor GoT, did we perform any post-measurement optimization of beacon positioning. In both cases, beacons were uniformly spread across the room in altitudes of 4 to 6 m, as shown below in Figure 2a. We plot beacon positions on top of ROSmaps to illustrate the mix of environments around ground-truth positions, i.e., whether they are in an area cluttered by equipment, shelves, and so on.

#### 2.1.3. Reference System

The reference system used in this study is provided by a Leica TS16 Total-station [45]. It has an angular standard deviation (std) of $2.35\times 10^{-5}$ rad and a distance std of 1 mm +1.5 ppm. This amounts to a radial std of 1.04 mm over the length of the laboratory, and a tangential std of 0.94 mm over the same range. The location of the Pozyx and GoT beacons has been established by the reference system, at the centre of the receiving antenna. For the measurement campaign, the reference system is used in the two major cases, static and dynamic, as described below.

Static. The 2D coordinates of the ground-truth points on the floor are given by the total-station through measurements of horizontal and vertical angles and distances from all setups. The method employed is least squares. It provides a standard deviation of the measured angles of 0.001 degrees and of the measured distances of 0.002 m [45]. In order to measure all ground-truth points, the total-station needs to be set up in different locations in the laboratory to ensure LoS. Localization of beacons is performed at the point of the receiving antenna. The standard deviation of the estimated coordinates of beacons is less than 0.005 m [45]. The standard deviation of the estimated coordinates of the points on the floor is less than 0.002 m.Dynamic. The same reference system is used for the dynamic tests as in the static case. Positions of the object are logged with a frequency of 10 Hz. Before the dynamic test started, the position and orientation of the total-station was computed relatively to the beacons. The accuracy of the logged positions relative to the beacons is better than 0.010 m.

### 2.2. Physical Testing Facility and Beacon Placement

The testing facility comprises a 14 m × 40 m educational laboratory for industrial manufacturing (AAU Smart Lab). It is densely populated with various equipment such as educational production line, robot work cells, high-power laser cutter, as well as several desks chairs and metallic shelves. The contour of these obstacles can be seen using a ROSmap as in Figure 2.

A number of beacons for localization are placed close to the ceiling and walls; 8 for Pozyx and 14 for GoT, as shown in Figure 2a. Since the range of the US transmission in GoT is lower than the corresponding range of the UWB transmission in Pozyx, it was considered appropriate to use a greater number of GoT beacons. 100 ground-truth positions are shown in Figure 2b.

### 2.3. Methodology and Evaluation Metrics

The methodology of the measurement campaign is divided into a static and a dynamic part:The static part involves a tripod with 3 tags attached at 3 different heights. The lowest height of 0.3 m represents a typical height of indoor autonomous mobile robots, 1 m as the height of a production line or workers’ belt, 2 m as the height of an autonomous forklift or a worker’s helmet. Measurements of the tags are performed for 90 s for each ground-truth point presenting an aggregated result for the entire laboratory. Before the actual measurement campaign started, an extensive sensitivity analysis of the localization systems was performed in order to find the most suited system features for the AAU Smart Lab. The sensitivity analysis is summarized in Appendix A. The choice of a 90 s data acquisition period is the result of this analysis. Euclidean distances are used as evaluation metrics subject to further statistics, means and quantiles.A single ground-truth position with LoS conditions in both systems is selected for detailed static analysis. This shows, through auto-correlation analysis, that Pozyx measurements are subject to significant low-pass filtering. The detailed analysis for the selected position is used to predict and explain results for the dynamic part. Moreover, it suggests the use of inverse filtering in the dynamic part to ensure a fair comparison.The dynamic part involves attaching one tag on top of an indoor autonomous robot. The robot travels a predefined trajectory visiting almost all ground-truth points on the floor. The robot travels at an average speed of 0.5 m/s. Whereas the static evaluation is intended to show long-term average performance and reveal potential bias, the dynamic part is intended to show the performance when applied to a mobile robot and reveal potential short term error. Therefore, distance errors in X- and Y-directions are subject to high-pass filtering. Outputs of first-order high-pass filters with cut-off frequency 10 Hz are used as evaluation metrics for the dynamic case subject to statistical analysis. As suggested in the detailed analysis for a single selected ground-truth position, Pozyx measurements are subject to significant low-pass filtering. To account for this, we attempt the use of an estimated Wiener filter for both systems, which improves results for Pozyx, but not for GoT.

Both static and dynamic measurements are conducted under mixed LoS/NLoS conditions, i.e., both in open spaces, as well as cluttered environments surrounded by equipment, shelves, etc., characteristic for industrial environments.

### 2.4. Analysis of Secondary Evaluation Parameters

Apart from the immediate evaluation and comparison parameters regarding accuracy (static) and timeliness (dynamic), it is reasonable to mention other indirectly influential parameters as mentioned in Section 1.1, such as range, cost, maintenance burden, energy consumption and co-existence. A potential technology adopter/customer would to some extend consider the cost of installation and maintenance, as argued in [8,9,11,12].

The installation cost depends on the number and volume of installed equipment, as well as the cost per unit. We installed a number of fixed units/beacons according to directions given from suppliers, which resulted in 14 GoT (125 USD/unit) and 8 Pozyx beacons (400 USD/unit), respectively. This gives an installation cost for fixed equipment of ≈4000 USD (including a stationary PC and SW license) for the GoT system and ≈8000 USD (including gateway and SW licenses) for Pozyx, whereas mobile tag unit cost is 176 USD for GoT and 115 USD for Pozyx. Although Pozyx has a high initial cost, we find both systems within financial reach for most enterprises. Here, it should be noticed that the GoT system applied is a development version, and the cost estimates are obtained from the supplier GamesOnTrack.

Maintenance costs depend heavily on equipment lifetime. At the time of writing, we have not been able to obtain lifetime information for the two systems. Tag power consumption is 100 mA at 3.3 V for GoT and 180 mA at 3.3 V for Pozyx. For the GoT system, tag scalability is unlimited, as is also reported by Pozyx for the Enterprise version used in this evaluation. GoT uses 40 kHz US and 869 MHz radio signalling, whereas Pozyx applies UWB signalling in 500 MHz bands residing within 3–7 GHz according to IEEE 802.15.4. Whereas the UWB scheme in Pozyx offers a high level of resilience, some sensitivity to narrow band disturbance is expected for GoT. The US signalling in GoT opens for co-existence problems with robotic collision sensing equipment, which was also experienced during these studies.

Altogether, we find the two technologies comparable on secondary evaluation parameters furnishing the fairness of direct comparison on performance.

## 3. Results

### 3.1. Static Case—Aggregate Results

This section presents the analysis and results for the static tests carried out over the 100 ground-truth points. The estimated positions and standard deviations are shown, for an overview, in Figure 3 and Figure 4 for the 1 m height. The static 0.3 and 2 m height measurements are shown in Appendix B. Background data are available from https://doi.org/10.6084/m9.figshare.17429654 (accessed on 23 December 2021).

The (Euclidean) error distance averages calculated over the 100 ground-truth points at all three heights are between 0.34 and 0.65 m. The two systems are statistically indistinguishable from a static case perspective. This is a similar finding to the one in [38]. It is also statistically evident that the 2 m altitude improves average accuracy compared to the lower altitudes at 0.3 and 1 m. Error distance histograms for Pozyx and GoT for different altitudes are shown in Figure 5, Figure 6 and Figure 7. Aggregate error distance statistics are found in Table 2, whereas visual inspection of the histograms reveals high-error outliers in both systems.

The findings in the static case when aggregating over all 100 ground-truth points in condition of both LoS and NLoS reflect that the two systems are similar. That is, for 90% of the ground-truth positions at the 0.3 m and 2 m heights an accuracy of 1.3 m and 0.77 m is achieved, respectively. The mean is 0.34 m for both systems at the 2 m height, 0.65 m for Pozyx and 0.47 m for Got in the 0.3 m height case. Figure 8 illustrates the aggregated mean error distance for all heights and their standard deviations. The height of 1 m distinguishes the systems with an accuracy of 1.2 m for GoT and 2.1 m for Pozyx for the 90%-quantile and a mean of 0.69 m for Pozyx and 0.5 m for GoT. These results are reflected in Table 2. They also confirm the findings of [7], which disprove Pozyx’s claim for an accuracy of 0.1–0.3 m.

**Table 2 sensors-22-02927-t002:** Mean and median error distances for different altitudes. (Mean distance: E(d), Quantile: Q., Confidence interval: CI).

Altitude	E(d)/Std Pozyx	E(d)/Std GoT	Q50/0.9 CI Pozyx	Q50/0.9 CI GoT	Q90/0.9 CI Pozyx	Q90/0.9 CI GoT
2 m	0.3460/0.063	0.34/0.12	0.13/[0.11 0.15]	0.17/[0.12 0.22]	0.77/[0.38 1.34]	0.77/[0.48 1.1]
1 m	0.6986/0.14	0.5/0.14	0.16/[0.13 0.2]	0.19/[0.15 0.24]	2.1/[0.77 3.4]	1.2/[0.87 2.1]
0.3 m	0.6553/0.13	0.47/0.15	0.21/[0.16 0.26]	0.16/[0.11 0.2]	1.3/[0.8 2.2]	1.3/[0.75 2.3]

### 3.2. Static Case—LoS-Only Results

To accompany the static analysis of 90 s measurement intervals across all reference points, we present below the analysis of a selected reference position in open space to avoid significant influence from NLoS conditions. The purpose of this analysis is to characterize dynamic properties of both systems under LoS and static conditions in an attempt to predict or explain their effect on the dynamic results presented below. That is, the static analysis presented only 90 s aggregate statistics, heavily influenced by LoS/NLoS conditions for each ground-truth position. The analysis in this section attempts to avoid NLoS conditions, and studies in detail the behaviour of the produced time sequences from both systems. We consider a specific 90 s X-coordinate time series with mean subtracted for both technologies, as shown in Figure 9.

The variability of the position estimations show the difference between the two technologies. Over an interval of 90 s, the error varies within 0.4 m for Pozyx estimations compared to the 0.09 m for GoT as Figure 9 shows. We assume measurements have been subjected to low-pass filtering to reduce variance before being presented to the user, as shown in Figure 10.

Without knowing the details of such filtering, we assume a first-order AR-filter with unity DC gain, as given in (Equation 1).
(1)$$x_{n+1}^{f}+e_{n}^{f}={\hat{x}}_{n+1}^{f}=a{\hat{x}}_{n}^{f}+\left(1-a\right)\left(x_{n}+e_{n}\right)$$
where $x_{n}$ is the true coordinate (at time index *n*), $e_{n}$ is a zero mean white Gaussian unfiltered measurement error, ${\hat{x}}_{n}^{f}$ is the value presented to the user and *a* is the AR-filter parameter. We estimate for both Pozyx and GoT the single parameter *a* as a=0.981 and a=0.585 for Pozyx and GoT, respectively. Additionally, we estimate error variances $var\left(e_{n}^{f}\right)=\sigma_{f}^{2}\approx 10^{-2}$ m${}^{2}$ for Pozyx and $4\times 10^{-6}$ m${}^{2}$ for GoT. Hereof unfiltered measurement error variance can be obtained from Equation (Equation 2)
(2)$$var\left(e_{n}\right)=\sigma_{e}^{2}=\sigma_{f}^{2}\frac{1-a^{2}}{{\left(1-a\right)}^{2}}$$

For Pozyx, this results in an error variance of $\sigma_{e}^{2}=1.04$ m${}^{2}$ and for GoT $\sigma_{e}^{2}=1.6\;\times \;10^{-5}$ m${}^{2}$. These findings of a high variance of measurement error in Pozyx are consistent with [46]. If raw measurement noise is attributed to timer resolution of TDOA measurements, it corresponds to a resolution (see [47]) for Pozyx of $\frac{\sqrt{1.04\;\times \;12}}{C_{L}}=12$ ns and for GoT $\frac{\sqrt{1.6\;\times \;10^{-5}\times 12}}{C_{S}}=40\;\mu$s (where $C_{L}$ and $C_{S}$ are the speeds of light and sound, respectively). At first glance, this illustrates the very different technological challenges associated with radio- and sound-based localization, respectively. Next, 12 nS is somewhat contradictory to the applied pulse width of 0.16 nS announced in [48]. Estimated auto-correlations up to 4 s for both technologies are shown in Figure 11a, along with first order Auto-Regressive (AR) estimates (deduced from Formula (Equation 1)). It is known from technology providers that Pozyx runs with a default low-pass filter, whereas any filtering effect in GoT results stems from the iterative multi-lateration approach employed. As auto-correlations in Figure 11a show, Pozyx estimates are more heavily low-pass-filtered than their GoT counterparts. We decided to leave the low-pass filter in Pozyx turned on for the dynamic evaluation in the sequel, since this is the suggestion of providers for optimal tracking of mobile targets.

### 3.3. Position Tracking

When applying IPS for motion tracking, IPS measurements would surely be fused with other sensor-information such as, e.g., in a Kalman filter. In such a case, filters should match mobility characteristics, which would typically be of low-pass nature. In the following, we attempt to predict how IPS measurements from the two systems under investigation contribute to the performance of motion tracking. In order not to contemplate the details of the applied sensor fusion approach, we consider simple moving average filters with varying history length *N*. The N-sample average ${\bar{X}}_{N}$ is given in (Equation 3)
(3)$${\bar{X}}_{N}=\frac{\sum_{i=k}^{N+k}x_{i}}{N}$$
where $x_{i}$ is a measurement sample. The mean value estimate ${\bar{X}}_{N}$ has a variance $\sigma_{N}^{2}$ as given in Equation (Equation 4) (explained in Appendix C)
(4)$$\sigma_{N}^{2}=\frac{|Q_{N}|}{N^{2}}$$
where $|Q_{N}|$ denotes the sum of entries of the *N* by *N* auto-covariance matrix $Q_{N}$ defined in Equation (Equation 5)
(5)$$Q_{N}\left(i,j\right)=E\left(\left(x_{i}-\mu_{x}\right)\left(x_{j}-\mu_{x}\right)\right)$$
where $\mu_{x}$ denotes the common mean of $\left\{x_{i}\right\}$. We use Formula (Equation 4) to compare posterior standard deviations obtained for short interval averages, i.e., 0.5, 1, 1.5 and 2 s. Results shown in Figure 11b exhibit larger standard deviation (for short-interval samples) for Pozyx than for GoT. The error deviations are around 10 cm for Pozyx and sub-centimeter level for GoT. Additionally, notice the faster decay of standard deviation for GoT due to the faster decay of auto-correlation. We adopt moving average filtering from an agnostic viewpoint avoiding any speculation on the type of filtering approach applied in a practical situation. We generally assume that model-based filtering, such as Kalman filtering for, e.g., mobile robot navigation, will provide low-pass filtering adapted to the mobility behaviour of the robot, which can be represented as a weighted moving average. We, therefore, find the results shown in Figure 11b to be indicative of the performance of the technologies under investigation for mobile robotic applications.

Another finding is illustrated with the histogram of absolute errors in X-direction for Pozyx in Figure 12, where the log–log plot indicates power-tailed behaviour for errors between 60 cm and 2 m. The found power laws support the experimental findings of [49] regarding interference in IoT communication—considering Pozyx tags as IoT devices.

### 3.4. Dynamic Case Results

For the dynamic assessment and comparison, we define a laboratory-wide trajectory to be followed by a robotic ground vehicle (altitude 0.3 m). Positioning traces are continuously captured by Leica total-station, Pozyx and GoT. With the high relative precision (laser ranging) from the Leica total-station, we adopt this trace as the ground truth. XY-plots of all traces are shown in Figure 13.

The timely nature of the trajectory is made to mimic a relevant mobile intelligent robot serving transportation tasks of lightweight components with an average forward speed of 0.5 m/s in a factory.

#### 3.4.1. Filtering

For the dynamic assessments we let all error time-traces undergo pre-filtering by a first-order high-pass filter suppressing frequencies below 1/10 Hz and thereby, to a certain extent, static influences. This is done in an attempt to separate concerns related to static and dynamic error behaviour.

In a practical application, the combined influence of both static and dynamic errors has to be taken into account. The high-pass filter finally serves for de-trending measurements yielding wide sense stationary signals amenable to, e.g., frequency analysis, as applied below. An overview of the use of filtering in the dynamic error analysis is provided in Figure 14, where outputs 1,…,4 are subjects to subsequent statistics. Furthermore, Figure 14 shows how Wiener filters are tuned to match ground truth after pre-filtering.

As argued in the previous section, the two technologies under investigation seem to apply different filtering schemes to obtain their output results. To level this difference in accordance with the auto-correlation structure of the tested trajectory, each trace is subjected to Wiener filtering [50] with 4 s history estimated for optimal reproduction of the ground truth (Leica). Even though this might take more elaborate calibration procedures which are not practically feasible or likely, we find this approach more illustrative in terms of the full potential of the technologies. As above, we adopt Wiener filtering from an agnostic viewpoint, avoiding any speculation on the choice of filter for the eventual practical use.

Pre-filtered exemplary traces are shown with and without Wiener filtering in Figure 15 and Figure 16 for X- and Y-directions, respectively.

Pre-filtered traces in the frequency domain are shown with and without Wiener filtering in Figure 17 and Figure 18 for X and Y directions.

What is not immediately apparent from the exemplary traces in the time domain becomes apparent by inspecting power spectra; namely, that Pozyx holds a higher frequency error component (red), which is appropriately attenuated with the proposed Wiener filter, as shown in Figure 17a (red). (Power spectral densities are computed using the Matlab *pwelch* function with window length 500 and sampling frequency 10 Hz).

#### 3.4.2. Quantitative Error Assessment

Most readers would be accustomed to error assessment in terms of second moments, i.e., RMS error. The RMS results can be seen in Table 3. The Wiener-filtered RMS error shows little improvement from the unfiltered signal in GoT measurements. Both X-, Y-directions show a similar error and standard deviation. Pozyx measurements, however, benefit from Wiener filtering in both X- and Y-directions.

Statistics indicate power (heavy) tailed distributions or densities [51] of dynamic errors (see Figure 19), as seen in the static case also (see Figure 12). Powers of density tails (unitless) for absolute errors in X- and Y-directions (slopes) are found through inspection of log–log plots as 1.4 and 1.6 for Pozyx and 2.3 and 2.1 for GoT, respectively. Log–log plots of absolute error histograms are shown in Figure 19 along with a normal density with matched variance for comparison. Power tailed error statistics potentially yield excessively large variance in RMS estimates, since tail powers below 3 (heavy tails) yield infinite variance, when not truncated. Such excessive estimate variances and standard deviations are indeed seen for Pozyx results in Table 3, which makes drawing a distinction between technologies difficult.

Therefore, we resort to mean-absolute errors, with lower estimate variance as presented in Table 4, along with 90% error quantiles. Table 4 presents a statistically clear distinction between technologies with mean-absolute GoT errors approximately half the size of their Pozyx counterparts.

## 4. Conclusions and Future Work

In this paper, we have evaluated two positioning systems mainly targeted for indoor (GPS-denied) conditions. One system, Pozyx, is based on TDOA measurement with UWB radio signalling, whereas the other system, GoT, is based on TDOA measurements with US signalling. We restrict the comparison to US and UWB and do not include BLE.

Measurements are conducted in a 14 by 40 m laboratory facility at Aalborg University furnished with various industrial machinery, thus mimicking a realistic industrial environment, with both open and cluttered areas. Both systems include static (beacons) and mobile (tags) components. Beacons are for both systems installed according to recommendations from their suppliers. It is noticed that the installation costs of Pozyx are somewhat higher than those of GoT, whereas the tag costs of GoT are higher. Altogether, it is concluded that the two systems are initially comparable and the comparison is therefore fair.

Evaluation is divided into two main parts; static and dynamic. The former reports the statistics collected for 100 reference positions in 3 reference altitudes for 90 s averages. The latter records position measurements for tags mounted on a mobile robot (30 cm altitude) following a reference trajectory covering the entire facility. Static reference points, mobile robot positions and beacon positions are measured as ground truth with surveyor precision, i.e., 3 mm. In the static case, mean error distances between 34 and 65 cm are reported for both systems, although Pozyx shows significantly greater variance in spite of the more intensive low-pass filtering of the former.

For the dynamic case, errors signals are high-pass-filtered to separate dynamic effects from static ones. Moreover, optimal Wiener filtering is applied to predict the performance when applied for mobile robot localization in a sensor fusion conjunction. For the dynamic case, average absolute errors are within 35 to 42 cm for Pozyx and within 19 to 21 cm for GoT.

Overall, both systems suffer effects of NLoS conditions, which is seen from the significant errors of the static analysis. The results obtained tell a different story to the study in [38]: while for the static case the results are similar between the two technologies, in the dynamic case the US technology performs better than the UWB technology given LoS and NLoS conditions. For Pozyx, a significant error component is added through measurement granularity stemming from the choice of signalling medium. GoT suffers from coexistence problems in the US domain, mainly caused by US proximity sensors in industrial mobile robots. This was experienced during experiments as the dynamic measurements for the GoT system had to be taken with a customized robot without US collision sensors. For both systems, we find a statistically significant impact on tag altitude to performance, i.e., “the higher, the better”; this can be explained by the fact that greater altitude leads to greater probability of LoS conditions.

We suggest for both systems to reduce range and increase the recommended density of beacons. This would increase the number of simultaneous measurements and allow detection of NLoS measurements and the resulting devaluation of particular beacons. This suggestion indicates a direction for future research. An additional finding observed is power tailed error distributions in various cases, which for now remains unexplained, but are believed to relate to similar observations from other studies regarding received signals strength. Finding an explanation for tail error distributions is postponed until future research. With power tailed error distributions, the use of non-linear filtering may prove valuable and a promising direction for future research. Recent studies report improved performance of BLE as an accurate IPS technology, however, still not entirely at the level of UWB. BLE studies so far are limited to smaller and less diverse environments, which makes a direct quantitative comparison with our results impossible. An obvious direction for future research is the evaluation of BLE on equivalent terms, as in our study.

## Figures and Tables

**Figure 1 sensors-22-02927-f001:**
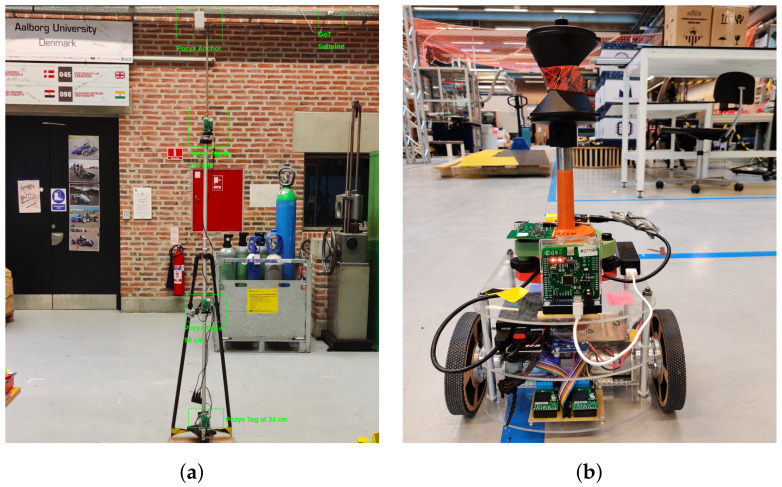
(**a**) Tripod for static measurements. Tags are placed on tripod at 3 heights. The background shows the beacons of the 2 positioning systems. (**b**) Mobile platform for dynamic measurements. Mobile robot carries both tags to be tracked. The optical prism (orange) is for the reference system.

**Figure 2 sensors-22-02927-f002:**
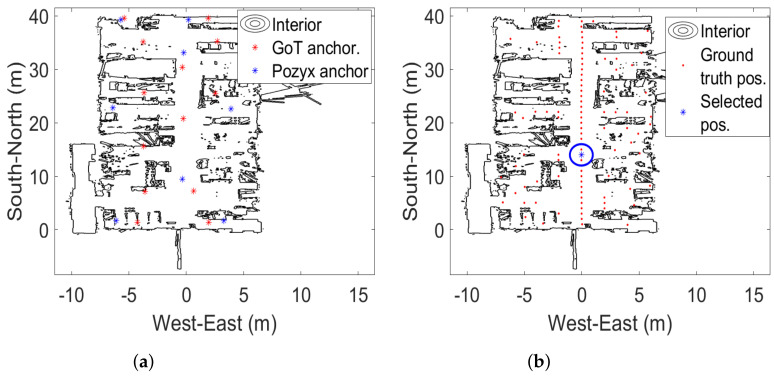
(**a**) Laboratory outline (ROSmap). Placement of Pozyx (blue) and GoT (red) anchors. (**b**) Placement of 100 ground-truth reference points (red). Ground truth point selected for detailed analysis (in blue circle).

**Figure 3 sensors-22-02927-f003:**
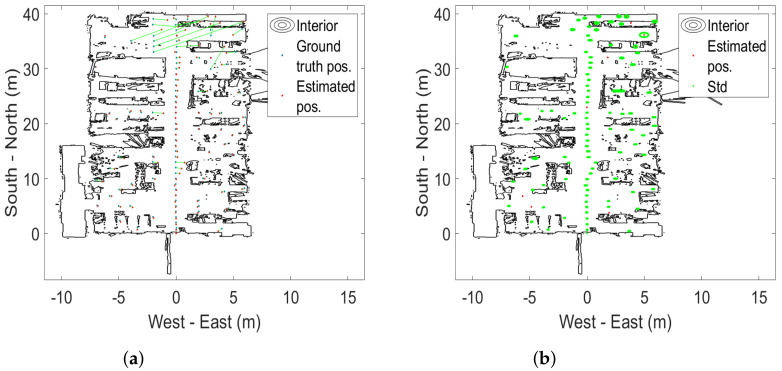
Static case results for Pozyx at 1m altitude. (**a**) Estimated positions from Pozyx (red) vs. ground-truth (blue) with connecting line (green). (**b**) Estimated positions from Pozyx (red) and standard deviation ellipsoids (green).

**Figure 4 sensors-22-02927-f004:**
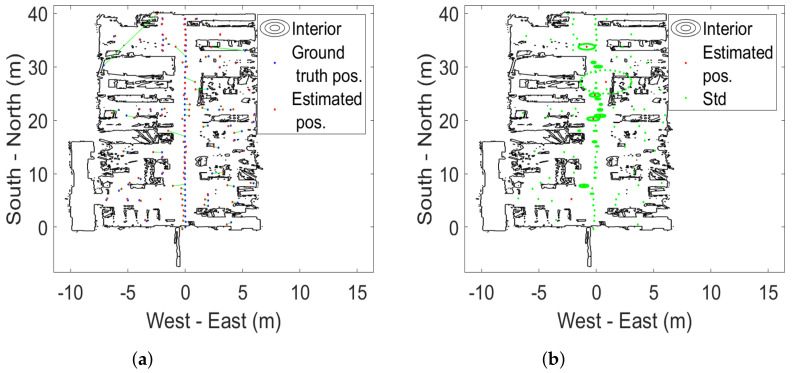
Static case results for GoT at 1 m altitude. (**a**) Estimated positions from GoT (red) vs. ground-truth (blue) with connecting line (green). (**b**) Estimated positions from GoT (red) and standard deviation ellipsoids (green).

**Figure 5 sensors-22-02927-f005:**
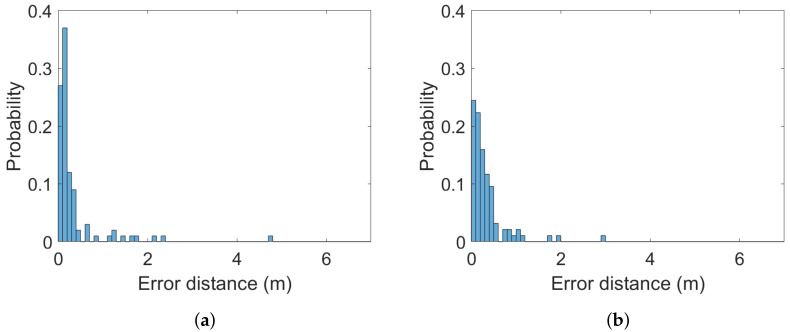
Histograms of error distances across all ground-truth points for (**a**) Pozyx and (**b**) GoT for 2 m altitude (bins: 0.1 m).

**Figure 6 sensors-22-02927-f006:**
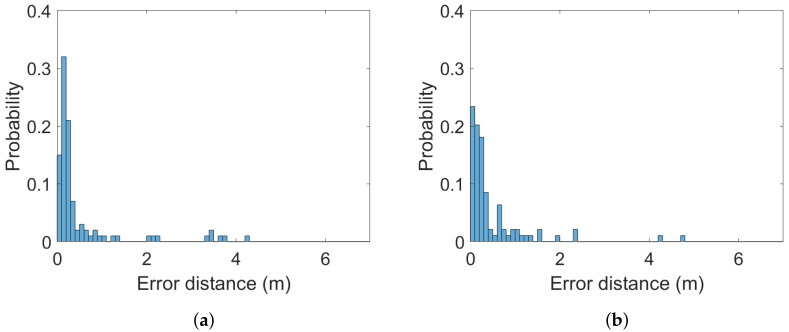
Histograms of error distances across all ground-truth points for (**a**) Pozyx and (**b**) GoT for 1 m altitude (bins: 0.1 m).

**Figure 7 sensors-22-02927-f007:**
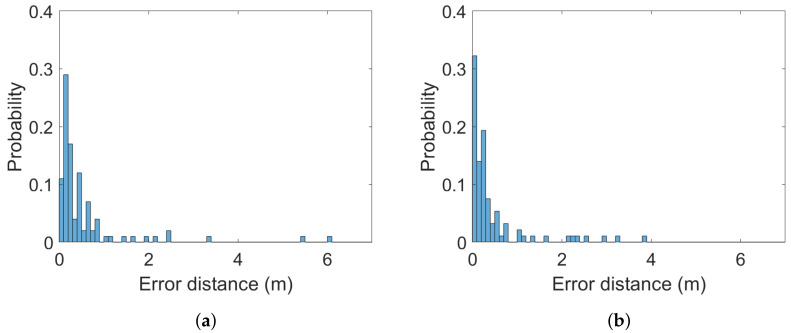
Histograms of error distances across all ground-truth points for (**a**) Pozyx and (**b**) GoT for 0.3 m altitude (bins: 0.1 m).

**Figure 8 sensors-22-02927-f008:**
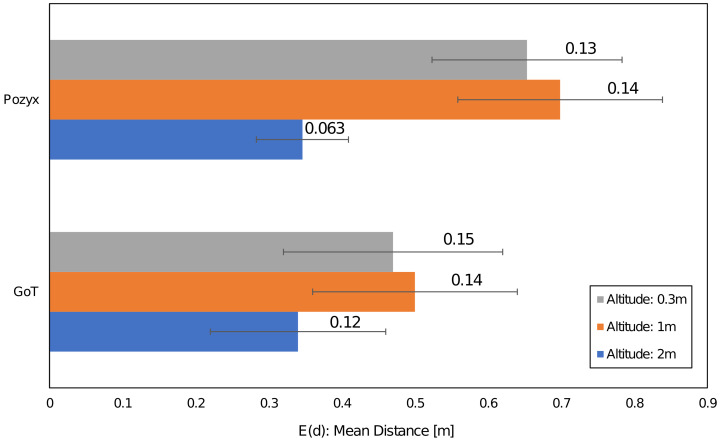
Aggregated mean error distance and standard deviation (as error bars) at 3 heights.

**Figure 9 sensors-22-02927-f009:**
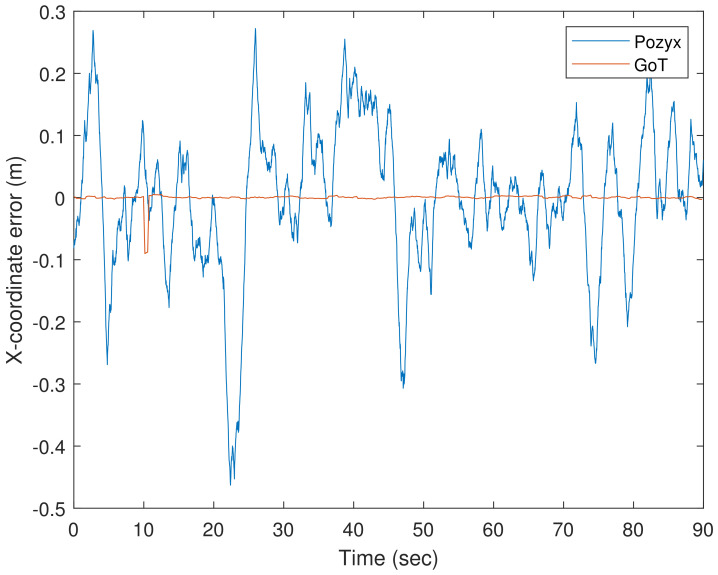
90 s of recorded X-direction errors from Pozyx (blue) and GoT (red) for ground-truth index 16 for the 1 m height.

**Figure 10 sensors-22-02927-f010:**
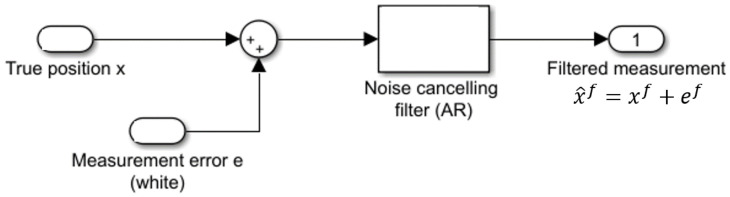
Assumed noise filtering of position measurements.

**Figure 11 sensors-22-02927-f011:**
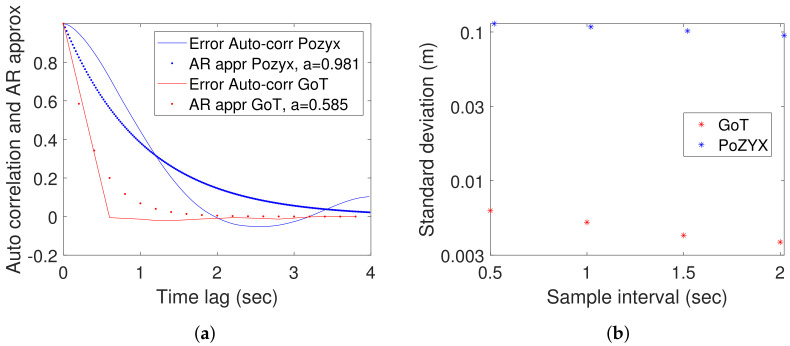
(**a**) Auto-correlation of X-coordinate for Pozyx and GoT along with AR-approximations. (**b**) Standard deviation after short term averaging (vertical axis is log scale).

**Figure 12 sensors-22-02927-f012:**
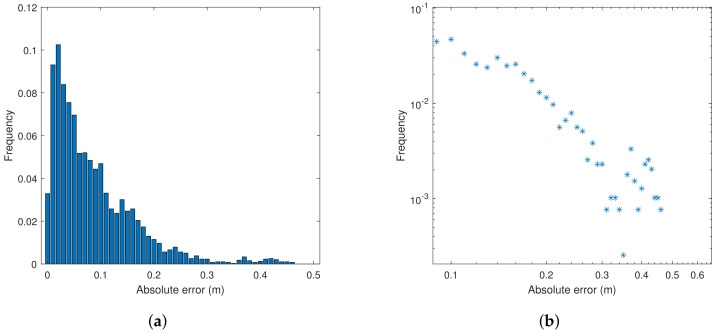
(**a**) Absolute error histogram for Pozyx (X-direction). (**b**) Log–log plot of absolute error histogram tail.

**Figure 13 sensors-22-02927-f013:**
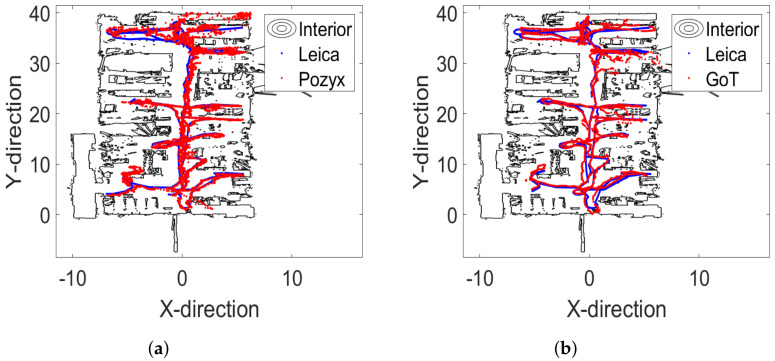
Two-dimensional traces for dynamic trajectory. (**a**) Laboratory outline (ROSmap) and Leica (blue)/Pozyx (red) traces. (**b**) Laboratory outline (ROSmap) and Leica (blue)/GoT (red) traces.

**Figure 14 sensors-22-02927-f014:**
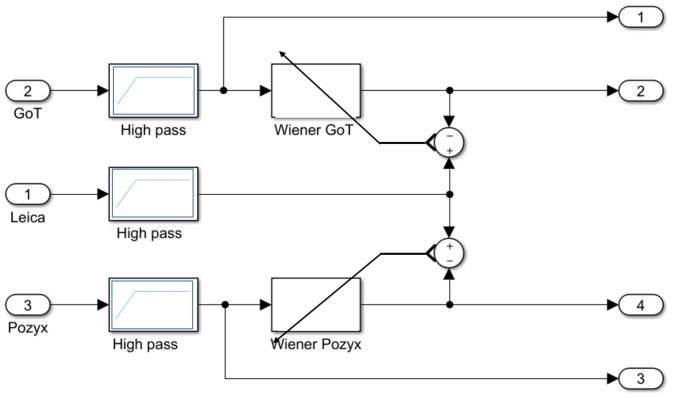
Use of high-pass pre-filtering and Wiener filtering during error analysis.

**Figure 15 sensors-22-02927-f015:**
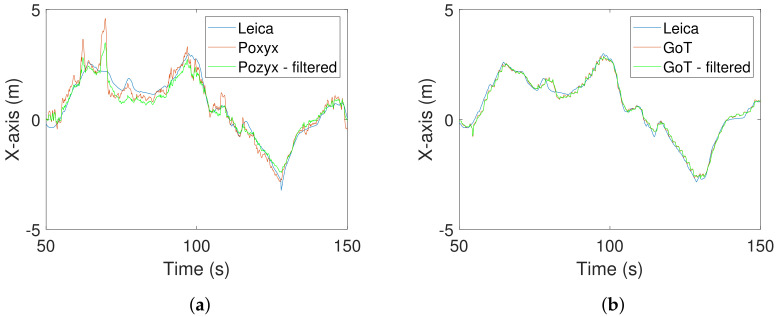
X-direction traces after pre-filtering with and without Wiener filtering. (**a**) Leica/Pozyx X-direction traces in time domain. (**b**) Leica/GoT X-direction traces in time domain.

**Figure 16 sensors-22-02927-f016:**
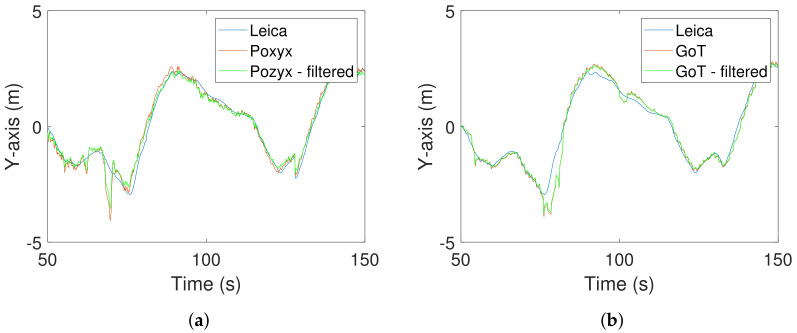
Y-direction traces after pre-filtering with and without Wiener filtering. (**a**) Leica/Pozyx Y-direction traces in time domain. (**b**) Leica/GoT Y-direction traces in time domain.

**Figure 17 sensors-22-02927-f017:**
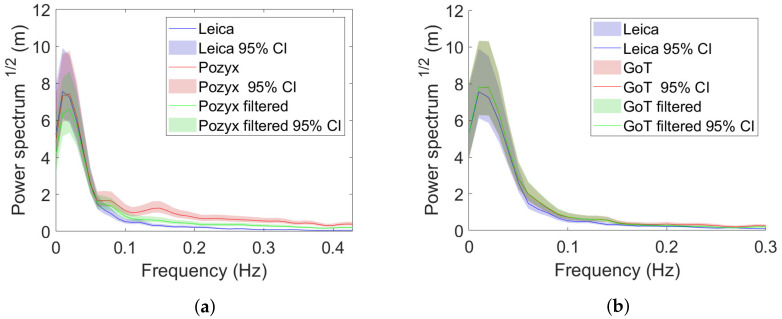
X-direction traces in the frequency domain after pre-filtering with and without Wiener filtering. (**a**) Leica/Pozyx. (**b**) Leica/GoT.

**Figure 18 sensors-22-02927-f018:**
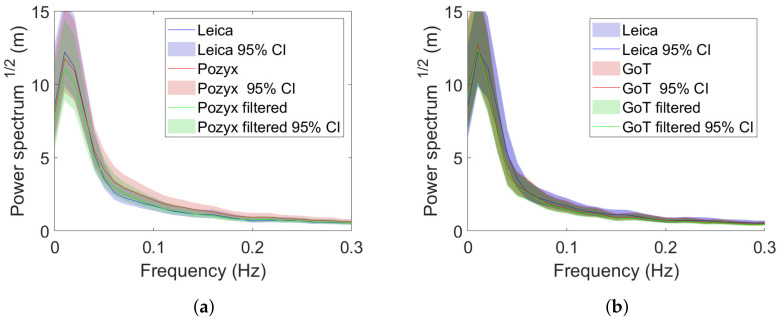
Y-direction traces in the frequency domain after pre-filtering with and without Wiener filtering. (**a**) Leica/Pozyx. (**b**) Leica/GoT.

**Figure 19 sensors-22-02927-f019:**
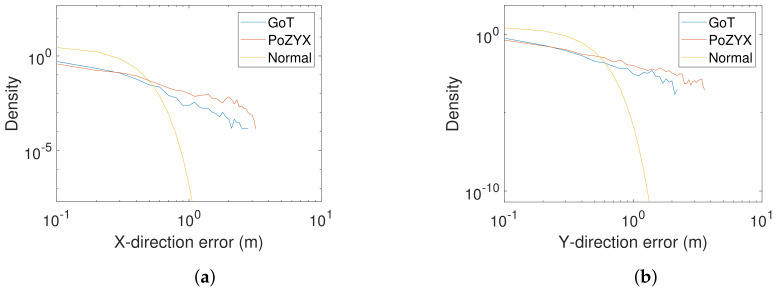
Log–log plots of absolute error for Pozyx and GoT, with scaled normal for comparison. (**a**) X-direction. (**b**) Y-direction.

**Table 3 sensors-22-02927-t003:** Filtered and unfiltered RMS errors for X- and Y-directions.

Direction	RMS Error Filtered [m]	Std [m]	RMS Error Unfiltered [m]	Std [m]
X-axis GoT	0.3289	0.013	0.3358	0.016
X-axis Pozyx	0.6224	0.3439	0.7139	0.7013
Y-axis GoT	0.3375	0.01	0.3437	0.013
Y-axis Pozyx	0.6172	0.3646	0.6599	0.4859

**Table 4 sensors-22-02927-t004:** Filtered and unfiltered mean-absolute errors for X- and Y-directions.

Direction	Mean-Abs Error Filt. [m]	Std [m]	Q90/95% CI	Mean-Abs. Error Unfilt. [m]	Std [m]	Q90/95% CI
X-axis GoT	0.2075	0.02	0.43 [0.3 0.48]	0.1992	0.02	0.42 [0.33 0.49]
X-axis Pozyx	0.3810	0.06	1.0 [0.58 2.0]	0.4202	0.07	1.2 [0.71 2.0]
Y-axis GoT	0.1984	0.02	0.43 [0.37 0.51]	0.1954	0.02	0.44 [0.37 0.51]
Y-axis Pozyx	0.3548	0.06	0.72 [0.61 1.3]	0.3779	0.07	0.78 [0.56 1.5]

## Data Availability

The data presented in this study are openly available in FigShare at https://doi.org/10.6084/m9.figshare.17429654 (accessed on 23 December 2021).

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
