# Peer review of "Evaluation and Comparison of Ultrasonic and UWB Technology for Indoor Localization in an Industrial Environment"

_sensors, 2022, doi:10.3390/s22082927_

Round 1

Reviewer 1 Report

The paper provides a comparative experimental study between two commercial positioning systems based on ultrasound and UWB technologies, respectively.

The paper is technically sound, and provides interesting results on the two technologies in a complex environment. Existing literature on the comparison between these two technologies is reviewed, providing a valuable input to researchers in the field. The experiment setup is also described in detail.

The paper is not highly innovative, since it focuses on performance evaluation, but it can provide insight on the performance of the systems in a challenging environment.

The paper needs however to be improved in the way the processing algorithms are introduced and used. In particular, the interaction between low pass filtering, high pass filtering and Wiener filtering introduced in Section 3 is rather hard to follow for the reader, and should be clarified.

Section 3 in general is often not clear. For example, the introduction of Equations (1)-(3) is not properly justified, and seems "disconnected" from the previous content.  

Finally, quality of presentation should be improved throughout the paper, both in terms of quality of English and in the quality of figures. All figures showing traces and estimated positions are rather hard to read, and in general most figures would benefit from a higher resolution.

The use of bloxplot figures would help conveying the different performance of the two systems in terms of variability of position estimation.

Author Response

Dear Reviewer 1

Thanks for most a valuable and constuctive review. We have tried to the best of our ability to accomodate all points raised. Detailled response to be found below (in red).

Best

Henrik Schiøler (corresponding author)

The paper provides a comparative experimental study between two commercial positioning systems based on ultrasound and UWB technologies, respectively.

The paper is technically sound, and provides interesting results on the two technologies in a complex environment. Existing literature on the comparison between these two technologies is reviewed, providing a valuable input to researchers in the field. The experiment setup is also described in detail.

The paper is not highly innovative, since it focuses on performance evaluation, but it can provide insight on the performance of the systems in a challenging environment.

The paper needs however to be improved in the way the processing algorithms are introduced and used. In particular, the interaction between low pass filtering, high pass filtering and Wiener filtering introduced in Section 3 is rather hard to follow for the reader, and should be clarified.

We support the reading by diagrams in figures 9 and 12

Also the accompanying text has been restructured to ease understanding

Section 3 in general is often not clear. For example, the introduction of Equations (1)-(3) is not properly justified, and seems "disconnected" from the previous content. 

We inserted a subsection "Motion tracking", where the rationale behind the study of averaging filters is presented.

Finally, quality of presentation should be improved throughout the paper, both in terms of quality of English and in the quality of figures. All figures showing traces and estimated positions are rather hard to read, and in general most figures would benefit from a higher resolution.

All figures have been resized to ease reading.

We checked the language all over with the use of Grammarly and a thorough read-through.

Reviewer 2 Report

I have some questions and comments about the job:

- The acronym GoT appears in the abstract and then throughout the text, but at no point is it defined as "GamesOnTrack".
- In section 2.1.1, the US device used is described as "development", but later in the cost estimate its price is included. If it is a commercial system it should be detailed exactly which model it is.
- Why is there a separation between paragraphs?
- Figure 1. The font of the axes is distorted, the aspect-ratio is incorrect. The font size is different in the sub-figure on the right and on the left. The caption should be changed, and sub-figures should be named as 1-a and 1-b. Right now the text is very confusing. Also some of the information that appears there should be added as a caption within each sub-figure.
- Figure 3. With the current size it is almost impossible to see the colors of the dots. Again the figure should have a legend. In the figure on the right most of the red dots are hidden by the standard deviation measure, they should be represented in another way. 
- Figure 4. Like the previous ones, the aspect-ratio on the axes is incorrect and the size of the dots is too small.
- Unify the acronym NLoS (sometimes it appears as "nLoS").
- In Figure 8-b you need to change the Y-axis values and use scientific notation.
- Before equation 1 you need a text explaining what comes next, otherwise you do not understand what the purpose of the formulas is.
- Figure 9 again has an incorrect aspect-ratio and the text of the axes overlaps.
- Figure 10, incorrect aspect-ratio.
- Figure 11-15, extremely small text font.
- There is no reference to the characteristics of the Leica system used, other than the name of the device. 

Author Response

Dear Reviewer 2

We thank you for a most valuable and constructive review. We have tried to the best of our ability to accomodate all the points raised. Detailled response to be found below (in red).

Best

Henrik Schiøler (corresponding author)

- The acronym GoT appears in the abstract and then throughout the text, but at no point is it defined as "GamesOnTrack".

We now specifiy under subsection "UltraSonic (US)" that GoT is in fact short for GamesOnTrack.

- In section 2.1.1, the US device used is described as "development", but later in the cost estimate its price is included. If it is a commercial system it should be detailed exactly which model it is.

The GoT system applied is a development version. The price given is indeed an estimate obtained from the supplier (GamesOnTrack). This is now explicated at the end of the section.

- Why is there a separation between paragraphs?

Here we dont exactly know what is meant.

- Figure 1. The font of the axes is distorted, the aspect-ratio is incorrect. The font size is different in the sub-figure on the right and on the left. The caption should be changed, and sub-figures should be named as 1-a and 1-b. Right now the text is very confusing. Also some of the information that appears there should be added as a caption within each sub-figure.

Aspect ratios are now corrected and subcaptions are inserted.

- Figure 3. With the current size it is almost impossible to see the colors of the dots. Again the figure should have a legend. In the figure on the right most of the red dots are hidden by the standard deviation measure, they should be represented in another way.

All figures have been resized to ease reading.

Legends are added.

We present data in this format to give the reader an overview and not to facilitate exact data dissemination. Exact data can be found in the repository: FigShare  https://doi.org/10.6084/m9.figshare.17429654. This is now made clear in the text.

- Figure 4. Like the previous ones, the aspect-ratio on the axes is incorrect and the size of the dots is too small.

See previous response

- Unify the acronym NLoS (sometimes it appears as "nLoS").

Corrected

- In Figure 8-b you need to change the Y-axis values and use scientific notation.

Corrected

- Before equation 1 you need a text explaining what comes next, otherwise you do not understand what the purpose of the formulas is.

We inserted a subsection "Motion tracking", where the rationale behind the study of averaging filters is presented.

- Figure 9 again has an incorrect aspect-ratio and the text of the axes overlaps.

Corrected

- Figure 10, incorrect aspect-ratio.

Aspect ratio has been corrected

- Figure 11-15, extremely small text font.

Fonts have been resized to ease reading

- There is no reference to the characteristics of the Leica system used, other than the name of the device.

Reference to the Leica product page from which specs can be downloaded is inserted. Basic specs are given in the text.

Reviewer 3 Report

The paper contains a potential valuable contribution. The subject is within the scope of the journal and the objective of research is well stated. However, some clarifications about the underlying hypothesis / scope as well as additional experiments clarifications are needed.

In the opinion of this Reviewer the manuscript deserves to be published once the Author takes into account the raised issues. A big rework is needed.

Introduction / Literature review

  1. The research scope is clear as well as the literature review. Anyway, the authors should better highlight the innovative aspects of their work in the manuscript.

What are the advantages / findings in the proposed paper, which are not covered by other studies/reviews?

  1. Row 160: check the acronym AoAAoD. This reviewer is not sure it is a new merged technique. They are two different approaches.
  2. The TDoA used as keyword and introduced in section 2.1.1 is completely missed in the introduction. In order to improve the literature review this review suggest citing recent work about TDoA (e.g. https://doi.org/10.1109/TCSI.2020.2979347)

Materials and methods

  1. 1: what do represent “S-N” and “W-E” on the axis? Are they the cardinal points? Please clarify this point. Too many parentheses make the text not clear. “… (red)(right). (unit(m)).”
  2. 3: what are right and left columns? Do the authors mean figure? Please add the unity of the measures. Please add a legend for more clarity. The figure is too small.
  3. 4. Please see the consideration made for fig. 3.
  4. Row 355. It is not clear to this reviewer if the measures are done for 90s o if 90s is the sampling time as stated at row 360. Please clarify this point specifying in any case, the sampling frequency and if some elaboration on the data is made.
  5. Row 420: 0.3m, 1m and 2m are the approximated values of 0.34m, 0.98m and 1.98m? It is not clear.
  6. 7: I suggest adding a legend to the figure. Tow different y-axis could be used also.
  7. Row 454: please recall the formulas in the order as they appear.
  8. Row 466: the equation 2 is not present in the appendix C, it is explained in the appendix C.
  9. Row 472: the maximum value shown in the figure is about 3.16cm, the values from 10cm to 20cm are not shown at all.
  10. 4: please explain all the terms in the equation
  11. Rows 492-493: which unity of measure is the capital “S”? Furthermore check the formulas, “1.04 12” something is missing?
  12. 9:please check the unity of measure for the x-axis for the left part. The maximum error is about 200m, are you sure? Probably are they mm? For the right part of the graph, check the x-axis values. They are not read well.
  13. 10: it is not clear to me if the authors reported two different experiments or not. The Lieca plots seem to be different. Why didn’t they use the same trajectory for the comparison? I suggest to enlarge the figure and put all the traces in the same graph.
  14. 11: please see the same considerations as for fig. 10.
  15. 12: why the y-axis is cleaner? Please give a comment on that.
  16. 13: please check the unity of measure of the power spectrum (meters?). Finally, how do the authors calculate the frequency starting from the time?
  17. Row 452: the figure 15 is not an histogram.
  18. The conclusions are missing.

Minor

  1. The authors should check that all the used acronyms are well explained and not repeated (e.g. UWB).
  2. The authors should check that all the formuals are cited in order of appearance
  3. Extensive editing of English language and style required. The paper should be carefully rechecked.
  4. Please specify the unity of measurement.
  5. Please use the letters for the subfigures instead of “left” and “right”

Author Response

Dear Reviewer 3

We thank you for a most valuable and constructive review. We have tried to the best of our ability to accomodate all point raised. Detailled response below (in red).

Best

Henrik Schiøler (corresponding author)

The paper contains a potential valuable contribution. The subject is within the scope of the journal and the objective of research is well stated. However, some clarifications about the underlying hypothesis / scope as well as additional experiments clarifications are needed.

In the opinion of this Reviewer the manuscript deserves to be published once the Author takes into account the raised issues. A big rework is needed.

Introduction / Literature review

The research scope is clear as well as the literature review. Anyway, the authors should better highlight the innovative aspects of their work in the manuscript.

What are the advantages / findings in the proposed paper, which are not covered by other studies/reviews?

Our contribution/innovation has now been explicated in subsection 1.2 of the introduction

Row 160: check the acronym AoAAoD. This reviewer is not sure it is a new merged technique. They are two different approaches.

AoAAoD is changed to AoA/AoD to ease the understanding that this is indeed a combined technique.

The TDoA used as keyword and introduced in section 2.1.1 is completely missed in the introduction. In order to improve the literature review this review suggest citing recent work about TDoA (e.g. https://doi.org/10.1109/TCSI.2020.2979347)

We have now used the TDoA keyword already in section 2.1.1, and extended the literature review by adding the suggested work.

Materials and methods

1: what do represent “S-N” and “W-E” on the axis? Are they the cardinal points? Please clarify this point. Too many parentheses make the text not clear. “… (red)(right). (unit(m)).”

Changed to South-North and West-East and subcaptions have been introduced decreasing the number of parentheses.

3: what are right and left columns? Do the authors mean figure? Please add the unity of the measures. Please add a legend for more clarity. The figure is too small.

All subfigures are now indexed (a),(b),(c),… The figure has been resized to ease reading.

  1. Please see the consideration made for fig. 3.

See response for figure 3.

Row 355. It is not clear to this reviewer if the measures are done for 90s o if 90s is the sampling time as stated at row 360. Please clarify this point specifying in any case, the sampling frequency and if some elaboration on the data is made.

We changed this text to avoid confusion. Now the term data acquisition period is used for the 90 sec period.

Row 420: 0.3m, 1m and 2m are the approximated values of 0.34m, 0.98m and 1.98m? It is not clear.

We now use everywhere the rounded altitudes 0.3m, 1m and 2m to prevent confusion.

7: I suggest adding a legend to the figure. Tow different y-axis could be used also.

A legend has been inserted. We prefer to keep the y-axes for Pozyx and GoT common - to ease comparison.

Row 454: please recall the formulas in the order as they appear.

We rearranged the text explaining the use of AR and moving average filters, so equations are now mentioned in the order they appear.

Row 466: the equation 2 is not present in the appendix C, it is explained in the appendix C.

Corrected.

Row 472: the maximum value shown in the figure is about 3.16cm, the values from 10cm to 20cm are not shown at all.

The figure and the accompanying text have been corrected.

4: please explain all the terms in the equation

All terms are now explained in relation to equation (4).

Rows 492-493: which unity of measure is the capital “S”? Furthermore check the formulas, “1.04 12” something is missing?

Units for second has now been corrected to lower case.

A dot is inserted into formulas to explicate multiplication.

9:please check the unity of measure for the x-axis for the left part. The maximum error is about 200m, are you sure? Probably are they mm? For the right part of the graph, check the x-axis values. They are not read well.

Figures have now been corrected to proper axes values and readability.

10: it is not clear to me if the authors reported two different experiments or not. The Lieca plots seem to be different. Why didn’t they use the same trajectory for the comparison? I suggest to enlarge the figure and put all the traces in the same graph.

All measurements are made simultaneously on the same mobile platform. For technical reasons the GoT trajectory is missing 1m of the initial trajectory from approximately -6,5 to -5,5.

11: please see the same considerations as for fig. 10.

Still the Leica trajectory is the same for both cases. The difference displayed in the figure is small and below significance and is explained by the different sampling frequency in Pozyx and GoT.

12: why the y-axis is cleaner? Please give a comment on that.

We find that this is mainly a result of coincidence for the excerpt trajectory taken out for time plot.

13: please check the unity of measure of the power spectrum (meters?).

We show the root (^(1/2)) of the spectrum to get an output in meters to assist interpretation.

Finally, how do the authors calculate the frequency starting from the time?

Power spectral densities are computed using the Matlab "pwelch" function with window length 500 and sampling frequency 10 Hz. pwelch implements Welch's method for spectral density estimation. This is now explicated in the text.

Row 452: the figure 15 is not an histogram.

We changed the word histograms to the less specific histograms and refer to loglog density plots

The conclusions are missing.

Title of the last chapter is now conclusions and the summary part is extended.

Minor

The authors should check that all the used acronyms are well explained and not repeated (e.g. UWB).

We checked this carefully throughout. In the title we allow the acronym UWB without explanation, since we find it to be commonly known among target readers. It is explained in the first use in the text.

The authors should check that all the formuals are cited in order of appearance

Some text has been restructured to make formulas being cited in order of appearance.

Extensive editing of English language and style required. The paper should be carefully rechecked.

The language of the paper has been checked throughout with the use of Grammarly and carefully proof-read.

Please specify the unity of measurement.

We added units all over

Please use the letters for the subfigures instead of “left” and “right”

Subcaptions have been introduced all over the text.

Round 2

Reviewer 2 Report

Following the review, the text has improved considerably. However, I still have some doubts and comments:

- Now that the data are published I have been able to look at them. Why do some of the ground truth values show some records with a negative time stamp?
- As a recommendation for the future, perhaps the format chosen for the publication of the results is not the most appropriate one. csv or even Matlab files are more usual.
- Also as a recommendation, it would be good to have all the raw measurements, before doing any averaging.
- The text of the axes in some figures is still distorted, as in figures 2, 3, and 4.
- Figure 8 is not in vector format, so it needs to be exported with more resolution to look good when enlarged.
- I don't know why table A1 is now in the appendix. In my opinion it contains the most important information of section 3.1, and although some of its data are discussed in the text, I think it is of more interest than figures 5, 6, and 7, where at first glance it is difficult to draw conclusions.
- Line 481 is cut off.
- Figure 11 is still very confusing despite the changes. Why are bars used? Why do they all have a top limit at 3e-1? 
- Figures 13 and 19 are still with distorted text, when others like 15, 16, or 17 are with the correct proportion. However, in figures 15, 16, and 18 the image on the right overlaps the last value on the x-axis of the image on the left.

Author Response

Dear Reviewer 2

Once again we thank you for a most constructive review.

Detailed response to be found below in red.

- Now that the data are published I have been able to look at them. Why do some of the ground truth values show some records with a negative time stamp?
- As a recommendation for the future, perhaps the format chosen for the publication of the results is not the most appropriate one. csv or even Matlab files are more usual.
- Also as a recommendation, it would be good to have all the raw measurements, before doing any averaging.
- The text of the axes in some figures is still distorted, as in figures 2, 3, and 4.
Figure aspects have now been changed to not distort texts. This is a tradeoff between readability in the figure and text axpect ratio
- Figure 8 is not in vector format, so it needs to be exported with more resolution to look good when enlarged.

fig 8 is now vectorized. 

- I don't know why table A1 is now in the appendix. In my opinion it contains the most important information of section 3.1, and although some of its data are discussed in the text, I think it is of more interest than figures 5, 6, and 7, where at first glance it is difficult to draw conclusions.
We reinserted the table in the main text.

Line 481 is cut off.
Line 481 continues correctly in line 482 with the reference [46].

Figure 11 is still very confusing despite the changes. Why are bars used? Why do they all have a top limit at 3e-1? 
This is because the vertical axis is indeed logarithmic, so there is no obvious origo. 
We changed to figure into a plot with '*' instead of bar plot. Then there is no top level of bars to confuse the reader.
Matlab seems to have problems with y-axis tick, when resetting the font-size. This has been taken care of and the vertical axis values changed slightly. 
The accompagning text (l. 501-506) has been changed slightly to fit the new values and to call attention to the faster decay of the GoT std.

Figures 13 and 19 are still with distorted text, when others like 15, 16, or 17 are with the correct proportion. However, in figures 15, 16, and 18 the image on the right overlaps the last value on the x-axis of the image on the left.
Text size and aspect ratio have been corrected in figs 13 and 19.
Overlap in figs 15,16 and 18 is removed

Reviewer 3 Report

Authors have properly enriched their work, by addressing each comment in a suitable way. The paper turns out to be notably improved.

Author Response

Dear, Reviewer 3

We thank you for a most constructive review.

This manuscript is a resubmission of an earlier submission. The following is a list of the peer review reports and author responses from that submission.

Round 1

Reviewer 1 Report

The paper contains a clear description of experimental campaign to compared UWB and US localization technologies in Industrial Environment. For that, authors provided a analysis of SOTA and measurement campaign in their laboratory.

The paper is a solid work with good technical contents with a completeness and accuracy references. Some figures are not clear, labels are not readable. 

The SOTA analysis is limited to similar example of measurement campaign in laboratory. I suggest to compared the results obtained with an real industrial environment measurement campaign. I suggest to read: A. Martinelli, et al., "UWB Positioning for Industrial Applications: the Galvanic Plating Case Study," 2019 International Conference on Indoor Positioning and Indoor Navigation (IPIN), 2019, pp. 1-7, doi: 10.1109/IPIN.2019.8911746. 

The importance and timeliness of the topic addressed in the paper within its area of research is good.

Author Response

Dear Rev1.

We thank you for your most careful and constructive review.

Please find our response (in red) to all the points you suggested.

Best

Henrik Schiøler (corresponding author)

The paper contains a clear description of experimental campaign to compared UWB and US localization technologies in Industrial Environment. For that, authors provided a analysis of SOTA and measurement campaign in their laboratory.

The paper is a solid work with good technical contents with a completeness and accuracy references. Some figures are not clear, labels are not readable.

We updated figures 5,6,7,8 for improved readability.

The SOTA analysis is limited to similar example of measurement campaign in laboratory. I suggest to compared the results obtained with an real industrial environment measurement campaign. I suggest to read: A. Martinelli, et al., "UWB Positioning for Industrial Applications: the Galvanic Plating Case Study," 2019 International Conference on Indoor Positioning and Indoor Navigation (IPIN), 2019, pp. 1-7, doi: 10.1109/IPIN.2019.8911746.

We added the reference to our list of references and cited in SOTA.

The importance and timeliness of the topic addressed in the paper within its area of research is good.

Reviewer 2 Report

The authors of the article state the results of an extensive evaluation and comparison of two positioning  systems.  One system, Pozyx is based  on TDOA measurement with UWB radio signalling. The second system, GoT  is based on TDOA measurements with US signalling. Measurements are conducted in a  laboratory facility, thus mimicking realistic industrial environment. Altogether it is concluded that the two systems are initially comparable and the comparison is therefore fair. It is possible to compare these technologies, but it requires an appropriate approach in the processing of measurement signals. I consider the results reported by the authors to correspond to the conditions in which the experiment was performed. I assume that using a nonlinear filter, the results of measurements using UWB technology would be better. I recommend publishing the submitted paper.

Author Response

Dear Rev2.

We thank you for your most careful and constructive review.

Please find our response to all the points you suggested.

Best

Henrik Schiøler (corresponding author)

The authors of the article state the results of an extensive evaluation and comparison of two positioning  systems.  One system, Pozyx is based  on TDOA measurement with UWB radio signalling. The second system, GoT  is based on TDOA measurements with US signalling. Measurements are conducted in a  laboratory facility, thus mimicking realistic industrial environment. Altogether it is concluded that the two systems are initially comparable and the comparison is therefore fair. It is possible to compare these technologies, but it requires an appropriate approach in the processing of measurement signals. I consider the results reported by the authors to correspond to the conditions in which the experiment was performed. I assume that using a nonlinear filter, the results of measurements using UWB technology would be better. I recommend publishing the submitted paper.

Non linear filtering may indeed be an excellent idea in this regard. However we find it to be a topic for future publications and has used the suggestion as a guideline for future research.

We added a sentence on non-linear filtering as a suggestion for a direction of future research.

We applied the filter proposed by the solution provider. We have no detailed knowledge about the nature of this filter, but trust that it meets the highest level of experience and competence for signal processing within UWB based indoor localization. We apply a Wiener filter calibrated on ground truth to provide indicative/predictive results for adequate filter performance. This in indeed done to remove a specific observed error component of the UWB signal.

Reviewer 3 Report

Comments to authors

Thank you for the works on the indoor localization systems. The paper tries to compare two technologies for the localization for industrial environment; however, the style, organization, and analysis of the paper do not meet the requirement of Sensors journal in the reviewer’s thought. The reasons as below.

  1. Please reads the ‘Instructions for Authors’ and follows the style.
    1. Here is the address https://www.mdpi.com/journal/sensors/instructions
    2. Abstract is less than 200 words. Abstract does not contain reference as well as acronyms.
    3. No italic word except mathematical variables
    4. Improper table style
    5. Multiple figures (a), (b) -> Figure 1a, Figure 1b
    6. Use Equation (1), Equation (2)
    7. Etc.
  2. Define the word before use.
    1. Accuracy
    2. 90 s
    3. Auto-Regressive
    4. AR parameter a
    5. Auto-covariance matrix
    6. Figure ??
    7. Wiener filtering
    8. Density tail coefficients
    9. Etc.
  3. Title of this paper indicates the general evaluation and comparison for ultrasonic and UWB technology but actually performs action on two products Pozyx and GoT. You have to justify the connection between the technologies and products.
  4. You used many ROSmaps in the paper but I cannot see the valuable information from those. Too small and not specific.
  5. Why use filters in the analysis? Note that your paper title is the evaluation and comparison. If the products (Pozyx and GoT) execute the statistical signal processing inside, then the processing is also part of the evaluation. Or if the filter is for default mode, turn off the mode. If you want to suggest the algorithm, use independent paper.
  6. In Statistic Tests and Analysis section, why do you address the cost? The cost is important but it is not relevant in the section.
  7. E(d) is mean; but, std is standard deviation. Use the consistent notation.
  8. Also use consistent unit format. For example, 2m or 2 m.
  9. In Figure 5, use the same y range.
  10. In line 364, Figure 8a and Figure 8b should be Figure 7a and Figure 7b
  11. Justify Equation (2). Why you use the auto-covariance matrix for variance?
  12. I do not understand the sentence in line 375 and 376.
  13. Also, I cannot understand the paragraph from line 381 ~ 394.
  14. 4e-6, 16e-5m, what are those? Use the consistent notation for number representation.
  15. In line 439, what is the number unit? What is the meaning?
  16. Appendix A does not provide the analysis, just statement.
  17. Appendix B is just list of figures.

Above are the just few of the reasons. The overview of the related works is excellent. Thank you.

Author Response

Dear Rev3.

We thank you for your most careful and detailed review.

Please find our response (in red) to all the points you suggested.

Best

Henrik Schiøler (corresponding author)

Thank you for the works on the indoor localization systems. The paper tries to compare two technologies for the localization for industrial environment; however, the style, organization, and analysis of the paper do not meet the requirement of Sensors journal in the reviewer’s thought. The reasons as below.

    Please reads the ‘Instructions for Authors’ and follows the style.

        Here is the address https://www.mdpi.com/journal/sensors/instructions

We have read carefully the ‘Instructions for Authors’ and revised the paper accordingly.

        Abstract is less than 200 words. Abstract does not contain reference as well as acronyms.

Abstract is revised and is now 200 words.

        No italic word except mathematical variables

Corrected everywhere

        Improper table style

        Multiple figures (a), (b) -> Figure 1a, Figure 1b

Corrected everywhere

        Use Equation (1), Equation (2)

We revised the paper to not use implicit reference to equations and figures.

        Etc.

    Define the word before use.

        Accuracy

We searched and  found, that the term “Accuracy” is a commonly known and well defined term, i.e. as the “degree of absence of error”. Therefore, we find it not necessary to define it prior to use.

90 s

Changed to 90 second everywhere

        Auto-Regressive AR parameter a

A reference to formula (3) is made at the first mentioning. Formula (3) shows exactly the structure of a 1st order AR filter.

        Auto-covariance matrix

A reference for appendix C is inserted showing the definition of the auto-covariance matrix as well as proving formula (2).

        Figure ??

Text above figure 10 is revised and a reference to figure 10 is inserted as an illustration of the reference trajectory

        Wiener filtering

Reference to Wiener filtering is inserted at the 1st mention

        Density tail coefficients

A reference to Mitzenmachers tutorial on power tails and their generative models is inserted

        Etc.

    Title of this paper indicates the general evaluation and comparison for ultrasonic and UWB technology but actually performs action on two products Pozyx and GoT. You have to justify the connection between the technologies and products.

The products chosen are prominent examples representing technologies and it is assumed that they represent the highest quality of implementations.  We found multiple examples of related work where title indicates technologies and content is the comparison of solutions, e.g. reference [26] in our paper.

We changed the abstract to explain early that we in fact compare solutions as examples of technologies.

    You used many ROSmaps in the paper but I cannot see the valuable information from those. Too small and not specific.

We find the use of ROSmaps a convenient way of illustrating the mix of environments around ground truth position, i.e. whether they are in an area cluttered by equipment, shelves etc. This holds for the dynamic trajectory as well. A sentence explaining this is inserted before the first use of ROSmaps.

    Why use filters in the analysis? Note that your paper title is the evaluation and comparison. If the products (Pozyx and GoT) execute the statistical signal processing inside, then the processing is also part of the evaluation. Or if the filter is for default mode, turn off the mode. If you want to suggest the algorithm, use independent paper.

The imposed filter in Pozyx is there to reduce measurement noise. It is assumed by Pozyx to be the best setting for mobile robotics. We inserted a sentence in the paper to justify the decision  to use the built in filter of Pozyx. In figure 8 and formula (4) we indicate the performance of measurements without the built-in filter of Pozyx but with short term averaging. This is not to suggest any operative algorithm but to indicate a predicted performance under mobility.

    In Statistic Tests and Analysis section, why do you address the cost? The cost is important but it is not relevant in the section.

We removed any reference to cost in that section.

    E(d) is mean; but, std is standard deviation. Use the consistent notation.

Corrected everywhere

    Also use consistent unit format. For example, 2m or 2 m.

Corrected everywhere to the former

    In Figure 5, use the same y range.

corrrected

    In line 364, Figure 8a and Figure 8b should be Figure 7a and Figure 7b

Corrected 7a and 7b combined in fig 7 and the reference is corrected

    Justify Equation (2). Why you use the auto-covariance matrix for variance?

Derivation of formula (2) is inserted in appendix

    I do not understand the sentence in line 375 and 376.

The sentence has been changed to provide further clarification

    Also, I cannot understand the paragraph from line 381 ~ 394.

Extended the explanation to provide more context and clarification

    4e-6, 16e-5m, what are those? Use the consistent notation for number representation.

e is changed to E everywhere. At first use “(E-notation)” is inserted

    In line 439, what is the number unit? What is the meaning?

Unit of density tail powers is by definition log(Probability/meter)/log(meter)=log(1/meter)/log(meter)=

-log(meter)/log(meter)=-1 and thereby unitless (since probability is unitless). We inserted “(unitless)” as the first introduction of the tail powers.

The accompanying text has been modified to provide clarity with a reference to infinite variance for tail powers below 3

    Appendix A does not provide the analysis, just statement.

The title of appendix A has changed to: “Summary of the sensitivity analysis performed on the Pozyx system in AAU Smart Lab” to reflect better the content of the appendix. We inserted a sentence in section 2.2 “Methodology and Evaluation metrics” with reference to appendix A to explain the initial settings of the Pozyx systems and the choice of a 90 sec average.

    Appendix B is just list of figures.

Appendix B is included for completeness and allows the reader with further interest to investigate performance for altitudes 2m and 0.3m

Inserted a short introductory text before figures in appendix B

Above are the just few of the reasons. The overview of the related works is excellent. Thank you.

Round 2

Reviewer 1 Report

The new version of manuscript is clearer than previous version. The figures are readable.

Author Response

Dear reviewer 1

Thanks again for your most insightfull and helpfull comments assisting us to provide a manuscript meeeting the highest standards.

Best

Henrik Schiøler (corresponding author)

Reviewer 3 Report

I can see the improvement on your article. However, many critical mistakes are still existed such as Figure 8 caption GoT(??). Also please be kind to the reader. I understand the e-notation but usually we use the power of 10 notation in publication. If there is a figure, the author should fully explain about the figure. Also, if there is a definition, the author should fully show the equation and meaning even with reference. Now I understand the reason that the paper selects the two products for general comparison according to the explanation. But I cannot comprehend the addressing in the section 3.2 and section 3.3.1. The reviewer is the logic checker rather than the proofreader. In your paper, I was too much distracted by the mistakes and depletion. I know that the author collects and performs the great experiments in the paper but the presentation and analysis should be revised further for the publication in Sensors. There is no right on reviewer to determine the level of research but the reviewer can decide the level of logical flow. By the way, there are mistakes on Appendix C equations.

Author Response

Dear reviewer3

Thanks again for your most detailed and helpful comments assisting us in improving readability and general quality of our manuscript. We have considered every comment and modified the manuscript as highlighted below.

Best

Henrik Schiøler (corresponding author)

I can see the improvement on your article. 
However, many critical mistakes are still existed such as Figure 8 caption GoT(??). 
--caption of Figure 8 has been corrected

Also please be kind to the reader. I understand the e-notation but usually we use the power of 10 notation in publication.
--E-notation has been changed everywhere to power of 10 notation

 If there is a figure, the author should fully explain about the figure. 
--references to figure 8 have been corrected
--table 3 somehow dissappeared in the previous revision - this is now corrected

Also, if there is a definition, the author should fully show the equation and meaning even with reference.
--Equation (3) has been inserted defining the covariance matrix QN

 Now I understand the reason that the paper selects the two products for general comparison according to the explanation. But I cannot comprehend the addressing in the section 3.2 and section 3.3.1. 
--More explaining text has been inserted to assist the reader in following the reasoning behind analysis in section 3.2
--The introductory text for section 3.3.1 has been modified and extended to assist the reader in following the reasoning behind use of high pass filtering of error traces.

The reviewer is the logic checker rather than the proofreader. In your paper, I was too much distracted by the mistakes and depletion. I know that the author collects and performs the great experiments in the paper but the presentation and analysis should be revised further for the publication in Sensors.
 There is no right on reviewer to determine the level of research but the reviewer can decide the level of logical flow. 
--we fully agree and hope that we have met the requirements with the current revision

By the way, there are mistakes on Appendix C equations.
--The index "i" has been changed to "j" in appendix C